# Behavioral Adaptations in Tropical Dairy Cows: Insights into Calving Day Predictions

**DOI:** 10.3390/ani14121834

**Published:** 2024-06-20

**Authors:** Aqeel Raza, Kumail Abbas, Theerawat Swangchan-Uthai, Henk Hogeveen, Chaidate Inchaisri

**Affiliations:** 1International Graduate Program of Veterinary Science and Technology, Faculty of Veterinary Science, Chulalongkorn University, Bangkok 10440, Thailand; aqeelkhosa@yahoo.com (A.R.); drkumail.abbas@yahoo.com (K.A.); 2Research Unit of Data Innovation for Livestock, Department of Veterinary Medicine, Faculty of Veterinary Science, Chulalongkorn University, Bangkok 10330, Thailand; 3CU-Animal Fertility Research Unit, Department of Obstetrics, Gynaecology, and Reproduction, Faculty of Veterinary Science, Chulalongkorn University, Bangkok 10330, Thailand; theerawat.s@chula.ac.th; 4Business Economics Group, Wageningen University and Research, 6706KN Wageningen, The Netherlands; henk.hogeveen@wur.nl

**Keywords:** animal welfare, activity behavior, smart biosensor, transitional period, machine learning algorithm, tropical climate

## Abstract

**Simple Summary:**

This study investigated the challenge of monitoring the activity movement pattern of dairy cows during the transition period and forecasted the calving day on tropical dairy farms. This study used activity behavioral data from 298 before-calving and 347 after-calving Holstein Friesian cows, as well as machine learning models for predicting birth. This study demonstrates that the cows giving birth for the first time had a shorter rest period and an increased activity pattern. Additionally, this study found that machine learning models can predict the day of birth. These findings could help farmers improve management and enhance animal welfare during this critical period.

**Abstract:**

This study examined changes in the activity patterns of tropical dairy cows during the transition period to assess their potential for predicting calving days. This study used the AfiTag-II biosensor to monitor activity, rest time, rest per bout, and restlessness ratio in 298 prepartum and 347 postpartum Holstein Friesian cows across three lactation groups (1, 2, and ≥3). The data were analyzed using generalized linear mixed models in SPSS, and five machine learning models, including random forest, decision tree, gradient boosting, Naïve Bayes, and neural networks, were used to predict the calving day, with their performance evaluated via ROC curves and AUC metrics. For all lactations, activity levels peak on the calving day, followed by a gradual return to prepartum levels within two weeks. First-lactation cows displayed the shortest rest duration, with a prepartum rest time of 568.8 ± 5.4 (mean ± SE), which is significantly lower than higher-lactation animals. The random forest and gradient boosting displayed an effective performance, achieving AUCs of 85% and 83%, respectively. These results indicate that temporal changes in activity behavior have the potential to be a useful indicator for calving day prediction, particularly in tropical climates where seasonal variations can obscure traditional prepartum indicators.

## 1. Introduction

Dairy cows’ behavior serves as a valuable window into their well-being, offering comprehensive insights into their physiological state, health, and even affective states [1,2]. However, conventional methods often rely on subjective human observations, making them time-consuming, labor-intensive, and prone to inter-observer variability [3]. Addressing these limitations necessitates an automated, quantifiable, and precise system for dairy cow behavior monitoring, particularly for animal health and welfare. Sensing technology and smart biosensors offer significant potential for continuous, real-time monitoring of behavioral variations [2,4], reducing workload and veterinary costs while maximizing farm efficiency and profitability [5].

Several studies on dairy cows’ behavior during the transition period (TP) in temperate and subtropical climates have been reported [6,7]. The unique climatic characteristics of the tropics, characterized by high ambient temperatures and humidity [8], substantially influence behavior and productivity [9]. For instance, heat stress often causes increased restlessness and altered eating patterns compared to cooler climates [10]. This divergence underscores the critical need for targeted research in tropical environments to optimize dairy cow management and productivity under these challenging conditions.

Monitoring dairy cow behavior during the transition period (TP) is vital for identifying health and reproductive issues. Observing changes in activity, eating, rumination, and social interactions can signal potential problems [11,12,13,14,15]. In tropical climates, precise interpretation of these behaviors is essential for effective herd management and enhancing productivity and reproductive success. Additionally, analyzing movement patterns of dairy cows during the TP can indicate discomfort or stress, thereby improving management strategies for tropical dairy farming [16,17].

Machine learning (ML), a subset of artificial intelligence (AI), holds significant potential for making robust predictions in dairy farm operations by utilizing diverse data sets, including animal status, environmental factors, and management practices, which improve adaptability and efficacy [18]. However, the effectiveness of ML algorithms, primarily developed and tested in temperate and subtropical climates for predicting calving events through behavioral analysis [16,19], remains largely unproven in tropical settings. These algorithms, which detect behavioral changes like decreased eating and rumination [20] and altered activity patterns [16,21], signaling impending calving [22], are crucial for understanding health and reproductive issues [23]. Yet, their applicability to tropical conditions, which may induce different behavioral responses during the TP, is still uncertain. 

Our research addresses this crucial gap by comprehensively characterizing the behavioral patterns of dairy cows in Thailand during the TP under unique climatic conditions, while simultaneously adapting and validating ML algorithms for calving predictions in tropical environments. This dual approach facilitates a robust evaluation of the algorithms’ efficacy and generalizability within tropical settings, ultimately enabling the development of accurate and reliable prediction alerts for dairy farmers.

## 2. Materials and Methods

### 2.1. Animal, Housing, and Calving Management

Data from 347 Holstein Friesian dairy cows (lactation 1.5 ± 0.75, range 1–4, gestation length 279 ± 28 days) were collected from Sithichoke Dairy Farm in the Nakhon Ratchasima Province, Thailand, from July 2021 to March 2023. Within this cohort, 298 cows (189 animals in lactation No. 1, 69 animals in lactation No. 2, and 40 animals in lactation No. ≥3) had complete data covering the 14-day prepartum, while 347 (249 animals in lactation No. 1, 57 animals in lactation No. 2, and 41 animals in lactation No. ≥3) animals had complete data covering the 14-day postpartum.

The climatic conditions in Thailand are characterized as hot and humid, with a mean temperature of 27 °C and a mean relative humidity of 74%. The mean monthly temperature ranges from 24 °C in December to 30 °C (the highest temperature) in April. The temperature humidity index (THI) is lowest in winter (almost mean of 73) and highest in summer (almost mean of 80) [24]. Dairy cows were managed following standard farm protocols, including being moved to the calving pen three weeks before the anticipated calving date based on the last insemination dates and health records. The calving barn was an enclosure with rubber bedding, located adjacent to the lactating cow barn and within visual range of the calving area. The bedding was maintained as necessary, and it was thoroughly cleaned with fresh water following each calving event.

The dairy cow received a total mixed ration twice daily at 06:00–06:30 and 17:00–17:30. The description of feed formulation used for TMR, and the chemical composition of feed analysis is given in Table 1. The barn featured a completely open longitudinal side with natural sunlight and a dedicated feed bunk. Dairy farm personnel managed the supplementary artificial light in the barn to ensure adequate daytime illumination for eating activities and dairy cow behavior monitoring. Clean and fresh water was available 24 h a day. 

On-site farm staff continuously monitored the calving pens. Dairy cows exhibiting calving signs (the appearance of a water bag) were closely monitored. If no progress was observed in calving, a vaginal obstruction examination was performed by the farm manager and professionally trained dairy farm personnel. Calving events were classified as normal if animals required slight assistance from one person [25]. Calving events involving multiple personnel, calf repositioning, or surgical intervention were classified as dystocia [26,27]. Post-parturition, dam–calf contact was permitted for 5 to 10 min to facilitate calf cleaning and stimulation.

### 2.2. Definition of Parturition

In this study, the completion of parturition was the target variable for our predictive machine learning (ML) model, with calf expulsion marking the definitive parturition time [28]. Dry cows were moved to pre-calving pens three weeks before the expected parturition date. However, our predictive model was evaluated starting at 14 days before the onset of parturition. Throughout the study, dairy farm personnel manually recorded the calving events.

### 2.3. Data Preparation

#### Processing Sensor Data

The AfiTag-II biosensor (Afikim Ltd., Kibbutz, Israel) is a 3D accelerometer sensor designed for real-time monitoring of dairy cows’ postural behavior. This electronic device detects both dynamic (animal-induced) and static (gravity-induced) acceleration, translating physical animal movement into an output waveform [17]. Leveraging triaxial accelerometer technology, AfiTag-II discerns and classifies postural behavior predicted by animal activity. Therefore, to assess and monitor behavioral data, a triaxial accelerometer-equipped sensor encased in plastic housing is secured to the rear leg of each animal. This allows for the automatic acquisition of postural behavioral metrics such as activity, rest time, rest per bout, and restlessness ratio. The description of these behavioral metrics is given below in Table 2. The AfiTag-II sensor’s accuracy in monitoring these metrics has been validated by previous research [29]. Sensor attachment occurred in the month preceding each cow’s expected calving, approximately four weeks before expected calving, to monitor prepartum and postpartum behavioral changes within the one-week adaptation period preceding data collection. Raw behavioral metrics were transmitted via Wi-Fi to an office computer, where they were gathered daily and subsequently retrieved manually in a comma-separated value (CSV) format. These raw data were then subjected to cleaning, sorting, and further analysis.

### 2.4. Statistical Analysis

Descriptive statistics were performed using SPSS software (version 29.0.1, IBM Corp., Armonk, NY, USA). Outliers were identified and removed prior to analysis. Generalized linear mixed models (GLMMs) were carried out to perform both univariable and multivariable analyses. The GLMM was structured to explore the impact of lactation numbers (1, 2, and ≥3), days surrounding calving (−14 to 14), and the months of sensor measurement (1–12) on activity patterns of dairy cows. The model accounts for the non-independence of repeated measurements taken from the same animals by incorporating random effects. The formulation of the model is given below.
Yijk = β0 + β1 × lactation number (i) + β2 × days surrounding calving (j) + β3 × months of sensor measurement (k) + µi + ϵijk
where: 

Yijk: activity parameter for the i-th cow on the j-th day related to calving during the k-th months of measurement;

β0: intercept, representing baseline activity level;

β1, β2, and β3: coefficient for the fixed effects of lactation number, days surrounding calving, and months of sensor a measurement, respectively;

µi: random effect for the i-th cow, capturing individual variability among cows, assumed to follow a normal distribution N (0, $\sigma_{\mu }^{2}$);

ϵijk: residual error term for the i-th cow on the j-th day during the k-th month, assumed to follow a normal distribution N (0, *σ*^2^).

Separate GLMMs were developed for each dependent variable: activity, rest time, rest per bout, and restlessness ratio. For univariable analyses, lactation numbers (1, 2, and ≥3), months of sensor attachment (1–12, indicating month of recording), and days surrounding calving (−14 to 14 days) were included as independent effects, while cow ID served as a random effect. Estimates of the least squares mean and their standard errors were derived from the predicted values of the univariable model for each lactation number (1, 2, and ≥3), for each month of sensor attachment (1–12), and for days surrounding calving (from −14 to 14 days). The multivariable analyses included only statistically significant findings from the univariable analyses. The final multivariable model calculated predicted values and their 95% confidence intervals, including both upper and lower limits. Model fit was assessed using residual plots.

To explore significant variations in the predicted values of the final models within and across lactation numbers, pairwise comparisons were conducted using one-way analysis of variance (ANOVA) and Tukey’s Honestly Significant Difference (HSD) tests in R Studio version 4.0.0 [30]. These analyses, tailored for each lactation number, evaluated predicted values including activity levels, total rest time, duration of rest per bout, and restlessness ratio. The evaluations took into account factors such as lactation numbers (1, 2, and ≥3), the period surrounding calving (−14 to 14 days), and the month of sensor attachment (1–12). For assessing variance among lactation numbers during the calving period (−14 to 14 days), 29 distinct analyses of the predicted values were conducted. Likewise, to determine differences in lactation numbers across various sensor attachment periods (1–12), 12 individual analyses were performed. The normality of the residual distribution and the homogeneity of the variance were both evaluated and found to be satisfactory. A significance threshold of *p* < 0.05 was set for all tests.

### 2.5. Development of Calving Prediction Model

To facilitate reproducibility, the machine learning (ML) sample code, cleaned data, and other supplementary files for predicting calving days are available at the GitHub repository: https://github.com/AqeelRaza51214 (accessed on 7 June 2024). The ML algorithms, including random forest, decision tree, gradient boosting, Naïve Bayes, and neural network (multilayer perceptron), were developed to accurately predict the calving day. Random forest, an ensemble-supervised ML algorithm that is deeply rooted in decision tree classification (DTC), trains the predictor tree using m bootstrap samples from the training dataset [31,32]. To generate each predictor, the algorithm employs a random selection of subset features, thereby breaking down the correlation between features. The decision tree relies on a collection of decision rules, represented by branches, to classify the terminal nodes into specific events (calving and non-calving events) [16,33]. Gradient boosting sequentially integrates with the boosting decision tree to reduce the overfitting error and enhance the model’s predictive capability [34,35]. In contrast, the Naïve Bayes model works on Bayesian rules, where each feature holds an independent value, offering a unique and robust approach for a particular use [23,32,36]. On the other hand, the neural network (multilayer perceptron) uses a vector of real value as an input value, an output layer, multiple hidden layers, and an activation function to understand the complex nonlinear relationship from the training dataset [16,32].

An 80% subset of the observations, designated as the training dataset, was utilized to develop predictive models. The remaining 20% of the data was used to evaluate the performance of these models. Evaluation metrics such as sensitivity, specificity, positive predictive values (PPVs), negative predictive values (NPVs), accuracy score, and F2 score were used to evaluate model performance. Additionally, the area under the receiver operating characteristic (ROC) curve, which represents the probability of a model ranking a randomly chosen true positive event higher than a randomly chosen negative event, was used to assess each model’s performance. The ROC curve graphically depicts the true positive rate (TPR) against the false positive rate (FPR) at varying threshold levels. 

To predict the calving day (day 0), six features were extracted from postural behavioral data: activity, rest time, rest per bout, restlessness ratio, month of sensor attachment (1–12, record month), and lactation No. (1, 2, and ≥3). A Shapley Additive Explanations (SHAP) model was used to identify the most important features for predicting calving events.

### 2.6. Programming Packages

Data cleaning and imputation of missing values were performed using Python 3.12.2 (http://www.python.org accessed on 7 June 2024) with NumPy [37], Pandas [38], and Scikit-learn [37] add-on packages to develop and evaluate the aforementioned ML algorithms. Jupyter notebook was used as the code editor for all analyses, and the Matplotlib and Seaborn libraries of Python v3.12.2 and Microsoft Excel (version 2023) [39] were used for data visualization [40,41].

## 3. Results

### 3.1. Effect of Lactation Number on Behavior around Calving

Prepartum data were available for 298 dairy cows, of which 63.2% (*n* = 189) were in lactation No. 1, 23.2% (*n* = 69) were in lactation No. 2, and 13.4% (*n* = 40) were in lactation No. ≥3. Postpartum data were available for 347 lactations, of which 71.8% (*n* = 249) were in lactation No. 1, 16.4% (*n* = 57) were in lactation No. 2, and 11.8% (*n* = 41) were in lactation No. ≥3.

In the univariable analysis presented in Table A1, a significant association was observed between lactation groups and the behavioral patterns of dairy cows. This relationship was evident through marked differences in activity levels, total rest time, duration of rest per bout, and the restlessness ratio among various lactation groups. The differences reached statistical significance, with a threshold set at *p* ≤ 0.05. Animals in lactation No. 1 exhibited the highest mean activity levels (227.2 ± 2.8 min/day), which were significantly greater than that of lactation No. 2 (207.4 ± 5.0 min/day) and lactation No. ≥3 (206.8 ± 6.6 min/day, Table A1). Rest time followed a similar pattern, with animals in lactation No. 2 dedicating most of their time to resting (681.6 ± 9.4 min/day), followed by animals in lactation No. ≥3 (660.7 ± 12.6 min/day), and lactation No. 1 dairy cows (568.8 ± 54.4 min/day). These differences in rest time between lactation groups were also highly significant (*p* ≤ 0.05).

The mean rest per bout demonstrates a clear trend across lactation groups, increasing progressively with lactation numbers. Animals in lactation No. ≥3 had the longest average rest per bout (79.3 ± 2.4 min/day), followed by animals in lactation No. 2 (74.0 ± 1.8 min/day) and lactation No. 1 cows (65.7 ± 1.0 min/day, Table A1). Finally, the restlessness ratio mirrored activity patterns, with animals in lactation No. 1 exhibiting the highest restlessness ratio (4.1 ± 0.1), indicating greater discomfort than lactation No. 2 (2.8 ± 0.2) and lactation No. ≥3 dairy cows (2.9 ± 0.2). These findings reveal a strong association between lactation groups and postural behavior in dairy cows.

Animals in lactation No. 1 displayed increased activity and discomfort, along with a lowered rest duration and shorter resting bouts. Conversely, animals in lactation No. 2 and lactation No. ≥3 exhibited a shift toward reduced activity patterns and restlessness ratios while dedicating more time to resting and engaging in longer resting bouts.

### 3.2. Periparturient Activity Changes across Lactation Groups in Dairy Cows

Figure 1, Figure 2, Figure 3 and Figure 4 show significant variations in activity levels, rest time, rest per bout, and restlessness ratio during the days surrounding calving, encompassing a 14-day period before and after the parturition event. On the actual day of calving, all measured activity parameters underwent marked changes. This pattern indicates that the labor stage leading up to calving has a strong behavioral impact.

Activity levels around calving exhibited dynamic changes, as shown in Figure 1. In the fortnight leading up to parturition (−14 to −1 days), the activity level remained relatively stable across all lactation groups. All groups experienced a significant increase in activity levels on the day before calving, reaching a peak on the day following parturition (day 1). First-lactation dairy cows (lactation No. 1) displayed the highest peak in activity levels (340.7 ± 7.8 min/day), significantly surpassing those of lactation No. 2 (329.4 ± 18.6) and lactation No. ≥3 (322.9 ± 18.1 min/day). This suggests a higher level of agitation in first-lactation dairy cows compared to those with more maternal experience. All lactation groups experienced a gradual decline in activity postpartum, returning to their pre-calving state within two weeks. This return to baseline (195.9 ± 3.8 min/day for lactation No. 1, 184.31 ± 8.0 min/day for lactation No. 2, and 179.5 ± 10.0 min/day for lactation No. ≥3) illustrates the dairy cow’s adaptation to the physical and behavioral shifts following calving.

**Figure 1 animals-14-01834-f001:**
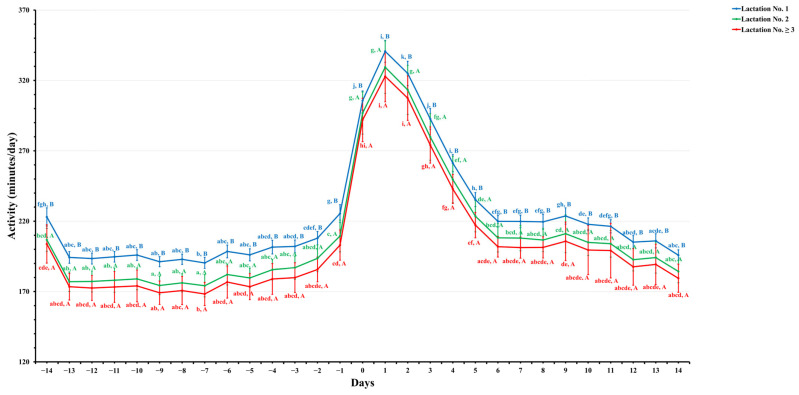
This figure depicts the estimated mean values with the standard error (SE) for prepartum and postpartum activity (minutes/day) for three lactation groups (1, 2, and ≥3). The estimated means were derived from predicted values obtained through a multivariable generalized linear mixed model for 14 days pre- and postpartum. Capital letters indicate significant differences between groups, while lowercase letters denote significant differences within groups, as determined by separate Tukey’s Honestly Significant Difference (HSD) tests. Activity (minutes/day) for each lactation group is color-coordinated (blue: lactation No. 1, green: lactation No. 2, and red: lactation No. ≥3). Data were collected from dairy cows equipped with an AfiTag-II biosensor (Afikim Ltd., Kibbtuz, Afikim, Israel).

Figure 2 shows a gradual decline in rest time for all dairy cows as parturition approached, starting from two weeks before calving (−14 to −1 days). On the day of calving, there was a significant difference between the groups. Animals in their first lactation had much shorter rest times (621.8 ± 17.0 min/day) than those in their second (709.5 ± 38.8 min/day) and third or greater (705.9 ± 43.0 min/day) lactation, with significance levels of *p* ≤ 0.05. This suggests a link between the lactation number and pre-parturition behavioral patterns.

Following parturition, all lactation groups showed an increase in rest time, yet these values remained slightly lower than those observed before calving, potentially indicative of postpartum adjustments and the initiation of milking routines. The rest times two weeks (14 days) after calving were 526.9 ± 9.1 min/day for dairy cows in their first lactation, 618.0 ± 19.0 min/day in their second lactation, and 619.0 ± 25.6 min/day for dairy cows in their third or subsequent lactation. This was in contrast to their longer rest times of two weeks (day −14) before calving, which were 688.9 ± 12.0 min/day, 781.4 ± 16.2 min/day, and 779.6 ± 23.2 min/day, respectively, as shown in Figure 2. This decrease in rest time after calving, despite a general trend towards recovery, may reflect the demands of the milking process and the associated physiological and behavioral changes in the postpartum period.

**Figure 2 animals-14-01834-f002:**
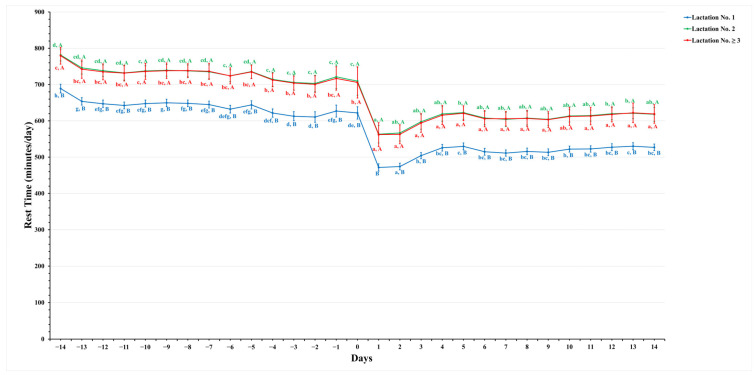
This figure demonstrates the estimated mean values with the standard error (SE) for prepartum and postpartum rest time (minutes/day) for three lactation groups (1, 2, and ≥3). The estimated means were derived from predicted values obtained through a multivariable generalized linear mixed model for 14 days prepartum and postpartum. Capital letters show significant differences between groups, while lowercase letters denote significant differences within groups, as determined by a separate Tukey’s Honestly Significant Difference (HSD) tests. Rest time (minutes/day) for each lactation group is color-coordinated (blue: lactation No. 1, green: lactation No. 2, and red: lactation No. ≥3). Data were collected from cows equipped with an AfiTag-II biosensor (Afikim Ltd., Kibbtuz, Afikim, Israel).

Rest per bout patterns around calving, as illustrated in Figure 3, revealed a significant decline leading up to the calving event, reaching the lowest values on the day of calving. Specifically, animals in lactation No. 1 had a rest per bout of 58.7 ± 0.9 min/day, while those in lactation No. 2 had 66.3 ± 2.1 min/day, and lactation No. ≥3 had the longest bouts at 74.6 ± 2.7 min/day. Notably, in all days surrounding calving, both pre- and postpartum, dairy cows in lactation No. ≥3 consistently exhibited the longest rest bouts than those in lactation No. 2, which in turn were longer than those in lactation No. 1.

Post-calving, all groups saw an increase in rest per bout, with lactation No. ≥3 exceeding pre-calving levels within a week and lactation No. 1 and 2 approaching their respective pre-calving levels. Importantly, after calving and the onset of milking, rest time did not revert to exact pre-calving figures, indicating a possible prolonged adjustment phase associated with the commencement of lactation and acclimation to the new milking regime. This observation highlights the lasting impact of calving on cow behavior and the recovery from associated stress, providing valuable insight into their postpartum adaptation.

**Figure 3 animals-14-01834-f003:**
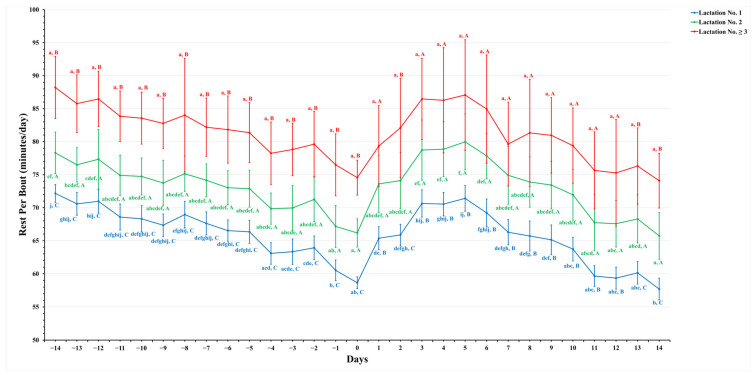
This figure shows the estimated mean values with the standard error (SE) for prepartum and postpartum rest per bout (minutes/day) for three lactation groups (1, 2, and ≥3). The estimated means were extracted from predicted values obtained through a multivariable generalized linear mixed model for 14 days prepartum and postpartum. Capital letters indicate significant differences between groups, while lowercase letters denote significant differences within groups, as determined by separate Tukey’s Honestly Significant Difference (HSD) tests. Rest per bout (minutes/day) for each lactation group is color-coordinated (blue: lactation No. 1, green: lactation No. 2, and red: lactation No. ≥3). Data were collected from dairy cows equipped with an AfiTag-II biosensor (Afikim Ltd., Kibbtuz, Afikim, Israel).

The restlessness ratio, which measures dairy cows’ restlessness and discomfort, displayed noteworthy changes surrounding parturition, as detailed in Figure 4. The restlessness ratio of all lactation groups remained stable until two days before calving, when it started to increase. On the calving day, the first-lactation dairy cows exhibited a significant increase in their restlessness ratio, reaching 5.6 ± 0.2, markedly higher than lactation No. 2 and lactation No. ≥3, which show ratios of 4.7 ± 0.4 and 4.6 ± 0.4, respectively (*p* ≤ 0.05). This suggests that first-lactation dairy cows exhibit more pre-calving restlessness than more experienced cows.

Following parturition, the restlessness ratio decreased, but it did not revert to the lower pre-calving levels within the two-week post-calving period. Instead, the restlessness ratio exhibited high values across all lactation groups (3.8 ± 0.2) for lactation No. 1, 2.8 ± 0.1 for lactation No. 2, and 2.7 ± 0.2 for lactation No. ≥3. This suggests that dairy cows are still very restless after calving, possibly because of the new demands of lactation and milking.

**Figure 4 animals-14-01834-f004:**
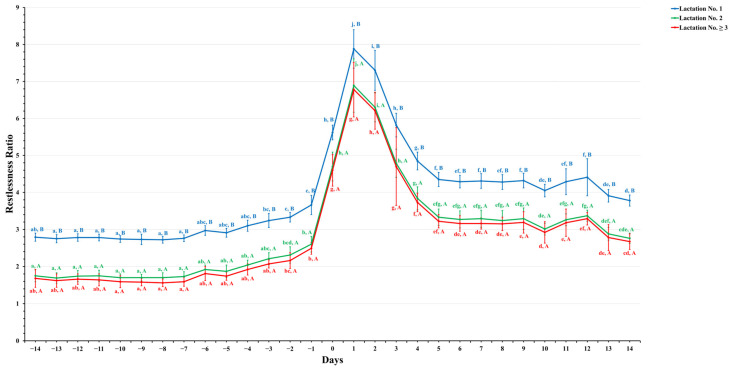
This figure illustrates the estimated mean values with the standard error (SE) for prepartum and postpartum restlessness ratios for three lactation groups (1, 2, and ≥3). The estimated means were generated from predicted values obtained through a multivariable generalized linear mixed model for 14 days, pre- and postpartum. Capital letters show significant differences between groups, while lowercase letters denote significant differences within groups, as determined by separate Tukey’s Honestly Significant Difference (HSD) tests. The restlessness ratio for each lactation group is color-coordinated (blue: lactation No. 1, green: lactation No. 2, and red: lactation No. ≥3). Data were collected from dairy cows equipped with an AfiTag-II biosensor (Afikim Ltd., Kibbtuz, Afikim, Israel).

### 3.3. Lactation Groups and Temporal Dynamics Influence Activity Patterns

In the multivariable analysis, an effect showing significance at a level of *p* ≤ 0.05 was detected between lactation groups (1, 2, and ≥3) and measurement months (1–12), influencing various activity parameters such as activity levels, rest duration, rest per bout, and restlessness ratio (Table 3). 

The present study recorded these data daily and combined pre- and postpartum data to analyze these data monthly for a year. Figure 5, Figure 6, Figure 7 and Figure 8 illustrate the seasonal behavioral changes in dairy cows across different lactation groups. The data for animals in lactation No. ≥3 in the month of April are missing. Lowercase letters on activity bars indicate significant monthly variations within lactation groups, while capital letters mark significant differences between lactation groups within the same month.

Throughout the year, dairy cows demonstrated marked variations in their activity levels. There was a significant increase in activity observed in June, with average activities recorded at 269 ± 5.9 min per day for first-lactation dairy cows, 325 ± 16.7 min per day for second-lactation dairy cows, and 241 ± 14.9 min per day for those in their third or subsequent lactations. These figures represent a substantial escalation compared to activity levels documented in other months.

For dairy cows in their second lactation, a distinct reduction in activity was noted in April, with an average activity of 165 ± 8.5 min per day. In contrast, dairy cows in their third or subsequent lactations experienced their lowest activity in July, with an average of 156 ± 4.9 min per day. This was in stark contrast to the decline in activity observed in first-lactation dairy cows during February, when activity fell to 199 ± 3.1 min per day. In June, second-lactation dairy cows showed a significant increase in activity, surpassing both the first-lactation and the third- and higher-lactation dairy cows. However, in not all months did the activity levels reach statistical significance in comparison between lactation groups.

**Figure 5 animals-14-01834-f005:**
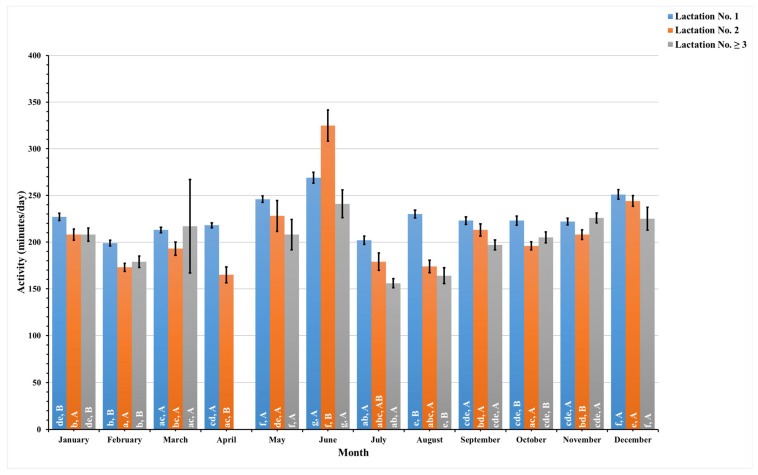
This figure shows monthly variations in the estimated mean values with standard error (SE) for prepartum and postpartum activity (minutes/day) across three lactation groups (1, 2, and ≥3) in dairy cows. The estimated means were derived from predicted values obtained through a multivariable generalized linear mixed model for months of measurement (1–12). Capital letters indicate significant differences between groups, while lowercase letters denote significant differences within groups, as determined by separate Tukey’s Honestly Significant Difference (HSD) tests. Data were collected from dairy cows equipped with an AfiTag-II biosensor (Afikim Ltd., Kibbutz, Afikim, Israel).

The present study meticulously recorded rest duration each month over a complete annual cycle, stratified by lactation numbers and quantified in minutes per day, as shown in Figure 6. The analysis revealed distinct fluctuations in rest duration throughout the year. Specifically, the shortest rest duration for dairy cows in their first and second lactations occurred in June, averaging 527 ± 8.7 and 534 ± 18.5 min per day, respectively. In stark contrast, the longest rest durations were observed in January for first-lactation dairy cows, with an average of 594 ± 7.5 min per day, and in April for second-lactation dairy cows, with an average of 749 ± 20.5 min per day. Third-lactation dairy cows, or those in subsequent lactations, displayed significantly longer rest durations over several months, with the longest durations occurring in May, averaging 987 ± 12.5 min per day. 

This pattern illustrates a synchronized behavioral trend among first- and second-lactation dairy cows, with the shortest rest period consistently seen in June. However, this trend was not as pronounced in third- or higher-lactation dairy cows, who did not show the same extent of reduction in rest duration in June, implying potential behavioral or physiological adaptations associated with more advanced lactation stages.

**Figure 6 animals-14-01834-f006:**
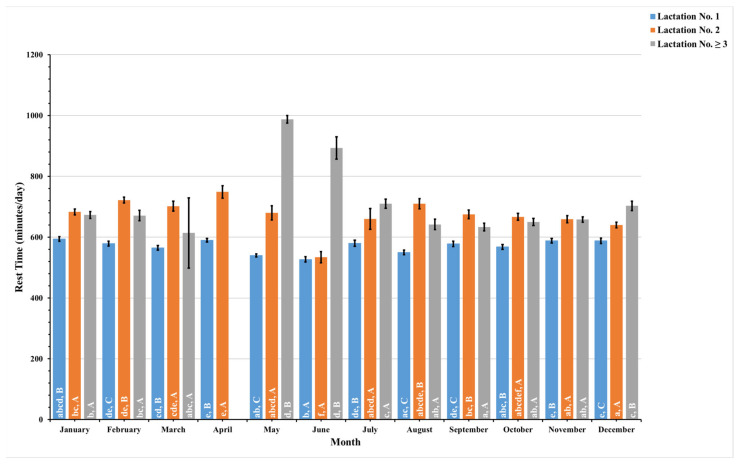
This figure shows monthly variations in the estimated mean values, along with the standard error (SE) for prepartum and postpartum rest time (minutes/day) in dairy cows with different lactation numbers (1, 2, and ≥3). The estimated means were derived from predicted values obtained through a multivariable generalized linear mixed model for months of measurement (1–12). Capital letters indicate significant differences between groups, while lowercase letters denote significant differences within groups, as determined by separate Tukey’s Honestly Significant Difference (HSD) tests. Data were collected from dairy cows equipped with an AfiTag-II biosensor (Afikim Ltd., Kibbutz, Afikim, Israel).

Figure 7 details the monitoring of rest per bout, a measure of the average duration dairy cows spend resting during each lying event, across the lactation groups throughout the year. Over the year, first-lactation animals experienced a decrease in rest per bout heading into June, bottoming out at 54 ± 1.0 min per day. Animals in lactation No. 2 displayed a similar pattern, with the shortest average rest per bout occurring in July, at 49 ± 2.6 min per day. Both groups saw an increase as the year progressed, with rest per bout duration peaking in October at 85 ± 1.8 min per day for first-lactation dairy cows and 88 ± 2.1 min per day for second-lactation dairy cows.

Animals in their third or subsequent lactations also showed this seasonal trend, with the least amount of rest per bout observed in June. As with the first two lactation groups, these cows exhibited an increase in rest per bout as the year advanced, culminating in the highest average in October, similar to their counterparts in earlier lactations.

**Figure 7 animals-14-01834-f007:**
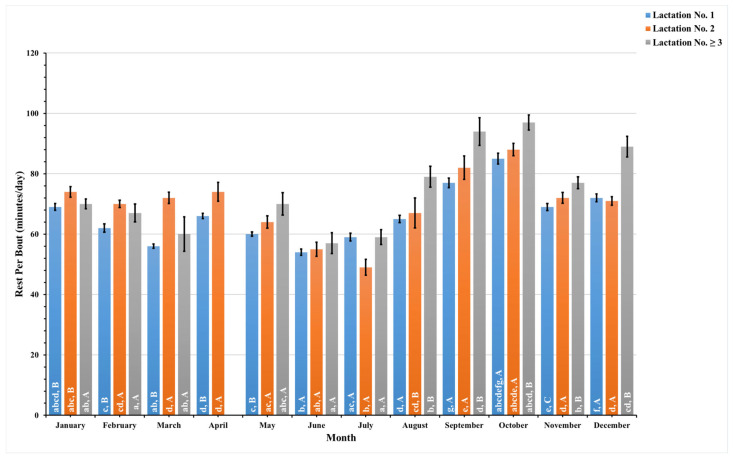
This figure illustrates monthly variations in the estimated mean values along with the standard error (SE) for prepartum and postpartum rest per bout (minutes/day) in dairy cows with different lactation numbers (1, 2, and ≥3). The estimated means were generated from predicted values obtained through a multivariable generalized linear mixed model for months of measurements (1–12). Capital letters denote significant differences between groups, while lowercase letters indicate significant differences within groups, as determined by separate Tukey’s Honestly Significant Difference (HSD) tests. Data were collected from dairy cows equipped with an AfiTag-II biosensor (Afikim Ltd., Kibbutz, Afikim, Israel).

The current study closely monitored the restlessness ratio, a measure of rest-related discomfort or stress, for dairy cows across three lactation groups throughout the year, as shown in Figure 8. First-lactation dairy cows consistently demonstrated a higher restlessness ratio each month when compared to dairy cows in the second and third, or higher, lactation groups. This persistent elevation peaked, notably in June, with younger cows exhibiting a restlessness ratio of 5.2 ± 0.2. In June, second-lactation cows experienced a similar peak, albeit slightly higher, with a restlessness ratio of 5.2 ± 0.3.

In stark contrast, cows in their third or subsequent lactations experienced their lowest level of restlessness in May, with a ratio of 0.9 ± 0.2. This lower restlessness ratio suggests that more mature cows may exhibit less rest-related stress or discomfort during this time, which may imply greater adaptability or a different set of needs that influence their behavior.

**Figure 8 animals-14-01834-f008:**
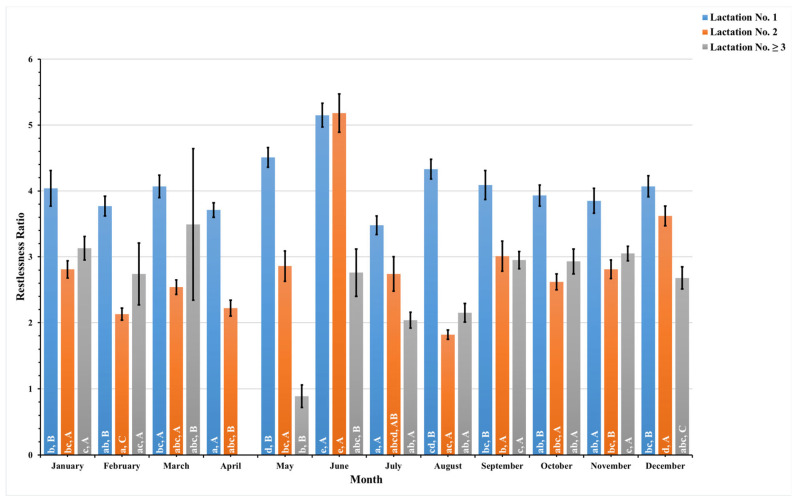
This figure depicts monthly variations in the estimated mean values along with the standard error (SE) for the pre- and postpartum restlessness ratio in dairy cows with different lactation numbers (1, 2, and ≥3). The estimated means were derived from predicted values generated through a multivariable generalized linear mixed model for months of measurements (1–12). Capital letters represent significant differences between groups, while lowercase letters indicate significant differences within groups, as determined by separate Tukey’s Honestly Significant Difference (HSD) tests. Data were collected from dairy cows equipped with an AfiTag-II biosensor (Afikim Ltd., Kibbutz, Afikim, Israel).

The analysis of dairy cows’ behavior across different lactation groups revealed that activity levels peak in June for all groups. Rest duration fluctuates seasonally, with all groups showing reduced rest per bout around mid-year and increased duration in rest per bout in October. First-lactation dairy cows displayed a consistently higher restlessness ratio throughout the year, suggesting higher stress sensitivity.

### 3.4. Machine Learning Model Evaluation

Table 4 presents the performance of five developed machine learning (ML) models for calving day prediction. The neural network (multilayer perceptron) had the highest specificity (98.9%), showing that it has a lot of potential for accurately detecting non-calving events. However, its sensitivity (40.0%) and F2 score (43.8%) suggest a potential risk of missing true calving events, as well as a less-balanced performance compared to other models. Random forest demonstrated a higher specificity (98.8%), effectively identifying false positive non-calving events. However, this comes at the cost of lowered sensitivity (40.0%) in capturing true calving events. The F2 score (43.7%) and accuracy score (95.2%) mark this trade-off balance. The decision tree and Naïve Bayes had the highest sensitivity (49.1% and 49.1%, respectively), which indicates these models were best at predicting calving days. However, their specificity was slightly lower (94.1% and 95.6%, respectively), and their F2 scores were slightly higher (45.6% and 47.5%). Meanwhile, gradient boosting had comparable specificity (98.8%) with random forest and neural network (multilayer perceptron), but they yielded the lowest sensitivity (34.6%) and F2 score (37.4%).

The receiver operating characteristics (ROCs) curve (Figure 9) showed that random forest and gradient boosting were the best at predicting calving day, with AUC values of 85% and 83%, respectively. Naïve Bayes and neural networks (multilayer perceptron) also demonstrated good predictive power, with AUC values of 82% and 81%, respectively. The decision tree model exhibited the lowest predictive capability, with an AUC value of 71%. Interestingly, models with higher F2 scores generally exhibited higher AUCs, suggesting their effectiveness in both identifying true calving events and minimizing false-positive events.

Using a Shapley Additive Explanation (SHAP) model, the present study evaluated the importance of features for calving day predictions in ML models. This model leverages concepts from game theory [42] and integrates seamlessly with the random forest regressor model in the present study [43] to illustrate the contribution of each input variable to the prediction of calving day models. Figure 10 presents the mean absolute SHAP values for each feature. Red dots on the higher end of the axis indicate a higher contribution toward the model’s predictions, while blue dots present on the lower end of the axis illustrate features with lower values that tend to decrease model performance. Positive SHAP values indicate a positive effect on the model’s ability to predict the calving day or specific instance [43]. On the other hand, negative SHAP values indicate that features may contribute to lower performance. The SHAP summary plot in the present study identified activity, rest time, and rest per bout as the most crucial features for developing an effective calving day prediction model. The SHAP summary plot illustrates each dot as a single SHAP value for a specific feature within a single data instance [42]. Notably, the clustering of dots reflects the strength of a feature’s interaction with the model’s outcomes [43]. It is important to concede that some classification ML models demonstrated lowered performances. The data file and code files are available at the GitHub repository: https://github.com/AqeelRaza51214 (accessed on 7 June 2024), which is available for further investigation. Future research should centralize feature engineering, hyperparameter tuning, and optimization, which have the potential to further refine and optimize these ML models and improve their effectiveness in generating accurate calving day alerts.

## 4. Discussion

### 4.1. Effect of Lactation Number on Behavior around Calving

This study analyzed prepartum and postpartum behaviors in 298 and 347 lactations, respectively, highlighting the substantial influence of lactation number on dairy cows’ behavior during calving. The distribution of lactation numbers indicates a majority in their first lactation, both prepartum and postpartum, followed by those in their second, third, or subsequent lactations. This demographic setup provides a solid foundation for assessing the influence of lactation experience on behavioral patterns.

The univariable analysis reveals a significant association between lactation number and various behavioral metrics, including activity levels, total rest time, rest per bout duration, and restlessness ratio. This association is statistically significant, illustrating distinct behavioral patterns across the lactation groups (Table A1). Notably, first-lactation dairy cows exhibited the highest mean activity levels (Table A1), possibly due to the combined stress of calving and adapting to a new lactating environment [12,44], which is further compounded by a lack of maternal experience [26,45]. Previous studies have shown that lactation affects dairy cows’ behavioral activity patterns as calving approaches [16,46,47]. The present study’s findings are consistent with previous studies [12,48], which reported elevated activity as calving approaches in first-lactation dairy cows. Additionally, dairy cows in their second lactation and subsequent lactations (lactation No. ≥3) displayed reduced activity compared to dairy cows in their first lactation, which is consistent with previous studies [16,26] that observed comparable results.

In contrast, dairy cows in their second lactation displayed the highest resting duration, followed closely by those in their third or subsequent lactation, with first-lactation dairy cows resting the least (Figure 2). This pattern suggests that higher-lactation dairy cows are more efficient at managing their energy management and rest prioritization, with more maternal experience to support upcoming milk production [46,49,50] as compared to first-lactation dairy cows. Additionally, resting duration decreased to the lowest levels as calving approached in all lactation groups compared to two weeks before calving (Figure 2). The resting duration of dairy cows in higher-lactation groups (lactation No. 2 and ≥3) contradicts the findings of [16,51], which indicated that first-lactation dairy cows rested more. This discrepancy might stem from age-related differences, variations in housing management, and physiological adjustments to the milking regime [12]. Furthermore, rest per bout durations increased with lactation numbers, with third or subsequent lactations showing prolonged rest bouts (Figure 3), indicative of better physical comfort and adaptation to the lactation cycle [52,53,54]. These findings are corroborated by [55], who noted a decrease in rest bout duration on the day of calving compared to the prepartum period.

The restlessness ratio further aligns these observations, with first-lactation dairy cows exhibiting the highest levels (Figure 4), indicative of higher discomfort possibly due to physical calving stress and psychological stress of transitioning to motherhood [6,12,45,54]. Interestingly, this elevated trend continues post-calving, while second- and subsequent-lactation dairy cows show less restlessness, suggesting they are more comfortable and better adapted. This study clearly delineates how lactation number impacts behavioral patterns around calving, highlighting the necessity of tailored strategies to support dairy cows, particularly first-lactation dairy cows, through this critical period. Understanding these patterns can guide better management practices, enhancing welfare and productivity across lactation stages. 

### 4.2. Periparturient Activity Changes across Lactation Groups in Dairy Cows

The present study offers a comprehensive analysis of behavior in dairy cows surrounding calving day, examining activity patterns, resting times, rest per bout, and restlessness ratios. Significant behavioral shifts were observed across all lactation groups around parturition, particularly in first-lactation cows, who exhibited increased activity and restlessness, along with reduced rest times and shorter rest bouts during the critical prepartum and postpartum periods. The limited sample size for each day surrounding calving constrained our analysis of interactions between days surrounding calving and lactation numbers, which likely differ across lactations. For instance, cows in later lactations might return to prepartum levels more quickly than those in their first lactation. Although we noted similar response patterns across various lactation numbers, suggesting consistent time-course shapes, these observations underscore the need for further studies with larger datasets to fully explore these interactions. However, this investigation into the complex interplay between physiological stress and adaptation across lactation stages provides key insights into dairy cows’ welfare during these pivotal times.

The present study noted an increase in activity beginning a few days before calving, which intensified after parturition, particularly among first-lactation dairy cows (Figure 1). Contributing factors include calving stress-related agitation [6], the physical demands of the initiation of milk production [51], and the challenges of integrating into an established herd [11,54]. Additionally, this study observed a gradual return to pre-calving activity levels within two weeks postpartum, indicating successful adaptation [11]. Interestingly, the present study’s findings slightly differ from the findings of [56], which indicated a quicker normalization of activity levels occurring nine days after calving, showing a recuperation from calving stress. Furthermore, these collective findings emphasize the importance of understanding and managing postpartum behavior to maximize cow comfort and welfare.

Changes in resting behavior, crucial for indicating post-calving comfort and stress, were also significant. All lactation groups showed a decrease in resting time immediately following calving (Figure 2), reflecting the universal impact of calving stress [11,26,51,57]. Notably, first-lactation dairy cows experienced a more substantial reduction in rest time, potentially leading to higher agitation [52] and vigilance during this novel experience of regrouping in the milking herd [57]. Despite the initial disruption, resting times did not return to prepartum levels within the two-week observation window (Figure 2), highlighting the sustained effects of calving stress and the new milking routine. These findings are consistent with previous studies [11,26,51,57], which reported similar observations of reduced resting time soon after calving.

The persistence of elevated activity and restlessness for two weeks postpartum underlines the challenges dairy cows face in adapting to post-calving life. The stress of milking initiation and integration into new or existing social groups [44,57] is significant. This prolonged adjustment period, characterized by notable changes in dairy routines and social hierarchies, particularly impacts first-lactation dairy cows [58]. Post-calving variations in rest per bout provide insights into dairy cows’ recovery from parturition. A peak in rest per bout durations 3–5 days post-calving indicates the effects required for the cow to recuperate, influenced by factors such as udder distention [59] and adjustments to new social and management regimes [54]. Additionally, higher-lactation dairy cows, facing increased energy demands, spend more time eating, which may lead to longer rest per bout durations [51,60]. However, animals in the first lactation had a smaller body size, which suggests an ease of transitioning from standing to resting and a shorter rest per bout duration than their counterparts. Previous studies [51,60,61] also reported a higher frequency and longer duration of rest per bout in higher-lactating animals, which is consistent with the present study’s findings.

These findings highlight the complex nature of dairy cows’ postpartum adaptation, influenced by factors such as calving experience [26], physiological stress [11], and the challenges of integrating into the milking routine [51]. Recognizing behavioral variance across lactation stages is essential for developing targeted management strategies that support dairy cows through this challenging period, ultimately enhancing welfare and optimizing productivity during the transition from pregnancy to lactation. Understanding these behavioral patterns helps improve management practices, boost welfare and production across the lactation phase, and benefit dairy operations’ overall health and effectiveness.

### 4.3. Lactation Groups and Temporal Dynamics Influence Activity Patterns

Environmental and climatic conditions significantly impact the behavior and well-being of dairy cows, serving as crucial indicators of their overall comfort and health [6,59]. This study explores the dynamic relationship between various lactation stages and the monthly behavioral patterns of dairy cows in Thailand, providing valuable insights into how these factors collectively affect dairy cows’ welfare. Notably, peaks in activity levels and restlessness ratio were observed in June for dairy cows in their first and second lactation groups (Figure 5 and Figure 8), coinciding with Thailand’s hot season. During this period, elevated temperature humidity index (THI) values, as a result of increased ambient temperature and humidity, correlate with the observed trends [62], suggesting a direct relationship between environmental stressors and the well-being of dairy cows.

Dairy cows in their third or subsequent lactation showed a lower level of restlessness in May (Figure 8), indicating potential adaptation or resilience to the climatic stressors experienced in earlier lactation stages. In contrast, first-lactation dairy cows exhibited a consistently elevated restlessness ratio throughout the year. The higher ambient environmental temperature and humidity during the summer likely contribute to this increased restlessness in first-lactation cows, possibly as a strategy to dissipate heat stress [62] and optimize thermoregulation [63] through increased body movement and blood flow. This study also identified a data gap for dairy cows in their third or higher lactation during April, potentially due to technical issues with monitoring equipment, external environmental stressors, or changes in management practices. This gap highlights the challenges of accurately capturing and interpreting dairy cows’ behavior, as well as the influence of external factors on their welfare.

The convergence of increased activity and reduced rest duration in June, particularly for dairy cows in the early stages of lactation (Figure 5 and Figure 6), underlines the significant impact of the hot and humid climate on dairy cows’ behavior. Conversely, as the year progressed into a cooler, though still humid, rainy season, we observed an increase in rest per bout duration (Figure 7), indicating a reduction in thermal stressors and allowing dairy cows to rest for longer periods. Dairy cows in lactation No. ≥3 exhibited less frequent but higher rest per bout patterns than those in their first and second lactations during the cooler months. This behavior reflects an adaptation strategy to the cooler months, characterized by lower ambient temperatures and humidity [63]. These environmental factors support longer resting bouts in higher-lactating animals, who often have larger body statures and heavier body weights, making frequent transitions from standing to resting more challenging [52]. Elevated rest per bout during cooler months also indicates higher levels of cow comfort [60], with previous studies [50] reporting similar seasonal variations in rest per bout patterns. 

This comprehensive analysis signifies the importance of understanding the complex nature of dairy cows’ behavior in response to environmental and climatic conditions [7]. It highlights the need for management strategies that are sensitive to the needs of dairy cows across different lactation stages and seasons [64]. Such strategies could include modifications to housing, feeding, and overall farm management to mitigate the effects of heat stress and optimize dairy cows’ welfare and productivity throughout the year. Future research should aim to elucidate the specific environmental, housing, and management factors contributing to the observed behavioral patterns. Direct THI measurements and in-depth analyses of housing and management practices will provide a better understanding of why dairy cows’ activity levels, rest durations, and restlessness ratios vary at different lactation stages and times of the year.

### 4.4. Calving Prediction

The exploration of machine learning (ML) models for predicting calving days in dairy cows demonstrates a significant stride towards harnessing technology to enhance dairy farm management and animal welfare [65]. Different ML models, such as neural networks, random forests, decision trees, Naïve Bayes, and gradient boosting, have shown different levels of effectiveness in predicting calving [16,21], which is an important event in dairy farming.

The present study’s results showed that the neural network model had the best specificity (Table 4), indicating its effectiveness in identifying non-calving days and minimizing false alarms. However, its relatively low sensitivity and F2 score indicate a potential shortfall in accurately detecting all true calving events. This gap emphasizes the importance of a balanced model that can both minimize false positives and ensure no calving event goes unnoticed [16,21].

Random forest and gradient boosting models emerged as notably effective, striking a meritorious balance between sensitivity and specificity. This balance is crucial, as it ensures that farmers can rely on the model to accurately predict calving, allowing for timely and necessary preparations and interventions without the burden of frequent false alarms [66]. The area under the curve (AUC) values from the receiver operating characteristics (ROCs) curve show that these models have a strong predictive ability for determining calving days. The decision tree and Naïve Bayes model prioritized sensitivity, reflecting a design choice to capture as many true calving events as possible, even at the expense of a higher rate of false positive alerts [16,67]. This approach is advantageous in scenarios where missing a calving event could lead to significant animal welfare issues or financial losses, highlighting the trade-off involved in model selection based on farm-specific properties [16]. On the other hand, gradient boosting’s lowered sensitivity indicates a need for further refinement. The model’s high specificity is valuable, but the ultimate goal is to develop a model that ensures no calving event is missed, highlighting the ongoing challenge in ML model development for calving prediction [21]. 

Our work demonstrated the potential for changes in activity patterns to predict calving days only. However, integrating environmental factors (such as THI), eating (chewing time and chewing bouts), rumination duration, body condition score (BCS), tail movements, and health records can optimize the ML algorithm’s sensitivity and accuracy. Technical limitations impeded the collection of these additional behavioral metrics. The inclusion of such data with activity patterns holds promise for generating effective and promising alerts for the day of calving and the day before calving. For example, the authors of ref. [3] were able to predict the day of calving with a higher sensitivity (77%), compared to this study’s 49.1% sensitivity, by using additional behavioral metrics such as the duration of rumination, changes in vaginal temperature, and period of rest time. Despite this inherent limitation, the current study’s features offer a valuable approach for calving day prediction in the absence of extensive behavioral monitoring. The present study used the Shapley Additive Explanation (SHAP) model to determine the most important features for predicting the calving day (Figure 10). This model highlights important behavioral metrics like activity levels, rest time, and rest per bout. These features are instrumental in predicting calving, with their variations providing crucial signals of impending parturition. The SHAP analysis underlines the significance of these variables, pointing to the need for models that can accurately interpret and leverage these signals.

We focused solely on the prediction of the exact calving day; however, ML algorithms can optimize calving prediction by generating alerts for both short and extended window sizes [16]. Extended window timeframes can substantially impact evaluation metrics, specifically the sensitivity and F2 score. Additionally, the extended window size (day before calving) and accurate predictions can provide farmers with extra time to oversee dairy cows and assist in calving complications, especially for large dairy herds [16,68], which have become prevalent in recent times. Furthermore, the day before calving alerts can assist farmers in strategically allocating their resources during the calving season and making suitable preparations for transitional dairy cows that may have faced any calving complications in the previous season [69].

Our work highlights the importance of understanding subtle changes in activity movement patterns during the TP, especially around calving in tropical dairy farming. The integration of the complexities of the ML model used for calving event identification in agricultural seminars, training sessions, and education programs for young veterinarians and farmers has immense potential. Understanding the complex relationship between ML models and model selection [70], feature importance [65], and sensitivity and specificity selection [16] is important for young veterinary generations and farmers. Therefore, seminars and hands-on training programs for farmers can integrate the findings of the current study to enhance their understanding of tropical dairy cows’ behavior. Future studies should focus on establishing a suitable balance between sensitivity and specificity, [16] and refine the features utilized in the present study. These tools can shape calving prediction alerts, provide effective teaching methods, and equip future generations of farmers and veterinarians with essential knowledge and skills to harness the power of artificial intelligence for optimal animal welfare and enhanced calving management.

Our results demonstrated a potential way to optimize calving management in tropical dairy farming systems. Using behavioral changes, the ability to predict the exact calving day could enable dairy farmers to implement target strategies to maximize cows’ comfort during this critical time. Timely alerts can provide farmers with an opportunity to prepare equipment and get trained personnel, potentially reducing the complications observed during this critical time window. Furthermore, timely intervention could reduce the chances of stillbirth and birth canal injuries, consequently affecting the decision to cull animals from the herd. This would improve both the welfare of the animals and the profitability of the farm. However, we recommend further studies to elucidate the complex relationship between the economic impact of timely calving alerts and tropical dairy farms.

## 5. Conclusions

This study aimed to identify distinct behavioral patterns in transitional dairy cows in tropical climates that could be a valuable tool for predicting calving days. The present study’s findings revealed significant changes in activity, rest time, rest per bout, and restlessness ratio across lactation groups and days relative to calving. First-lactation dairy cows exhibited the most nuanced changes, demonstrating increased activity and decreased rest time as calving approached. Machine learning models utilizing these behavioral metrics revealed promising accuracy for predicting calving days, particularly Naïve Bayes and decision tree algorithms. A potential limitation of our work is the absence of data regarding maintenance behavior (eating duration, chewing bouts, and rumination time), environmental factors (THI), and health records. The integration of these data points can further optimize these models’ performance. However, current research suggests that tropical dairy farming can use postural behavior as a valuable tool for calving day prediction.

## Figures and Tables

**Figure 9 animals-14-01834-f009:**
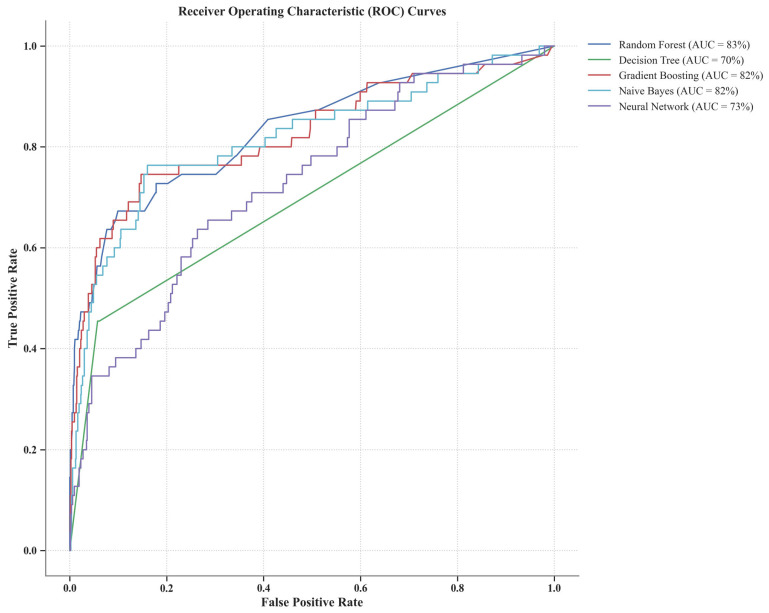
Receiver operating characteristic (ROC) curve for calving days prediction using classification machine learning algorithms in the test data set (80% of observations).

**Figure 10 animals-14-01834-f010:**
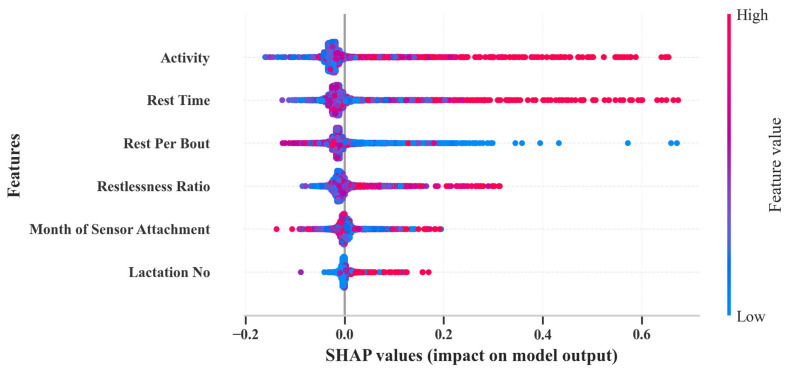
The SHAP summary plot indicates the features’ importance for predictions of calving days.

**Table 1 animals-14-01834-t001:** Description of feed formulation and chemical composition of TMR used in study.

Ingredients	Inclusion (kg/day)
Soya bean meal	1.1
Napier grass	30
Rice straw	4
Premix	0.14
Molasses	0.2
Selenium	0.08
Total	35.52
	Proximate analysis (dry matter basis)
Dry matter%	36.7
Crude protein%	16.7
Crude fat%	2.0
Crude fiber%	27.5
Ash%	26.1
	Detergent analysis (dry matter basis)
NEFL	27.7
ADF%	25.03
NDF%	39.26
ADL%	3.29
Cellulose%	21.74
Hemicellulose	14.23

NEF: Energy content (Mcal/kg); ADF: Acid detergent fiber; NDF: Neutral detergent fiber; and ADL: Acid detergent lignin.

**Table 2 animals-14-01834-t002:** Description of recorded behavioral metrics.

Behavioral Metrics	Description
Activity	The cumulative sum of the movement or physical activity displayed by the dairy cow per day.
Rest time	The cumulative duration the dairy cow spends lying down and resting per day.
Rest per bout	A measure of the average duration of continuous period during which the dairy cow remains lying down.
Restlessness ratio	A measure of how much the dairy cow is moving or shifting while it is lying down.

**Table 3 animals-14-01834-t003:** Multivariable analysis results of behavioral differences of activity, rest time, rest per bout, and restlessness ratio using generalized linear mixed models in relation to lactation numbers (1, 2, and ≥3), months of sensor measurement (1–12), and days surrounding calving (−14 to 14) with cow ID as a random effect.

Independent Variables	Activity (Minutes/Day)				Rest Time (Minutes/Day)				Rest per Bout (Minutes/Day)				Restlessness Ratio			
	β ± SE	*p*-Value	95% CI		β ± SE	*p*-Value	95% CI		β ± SE	*p*-Value	95% CI		β ± SE	*p*-Value	95% CI	
			Lower	Upper			Lower	Upper			Lower	Upper			Lower	Upper
Intercept	2017 ± 8.2	<0.0	185.7	217.7	641.5 ± 16.0	<0.0	610.2	672.8	77.7 ± 2.8	<0.0	72.1	83.3	2.9 ± 0.3	<0.0	2.3	3.5
Lact. 1	12.1± 7.2	<0.0	−2.0	26.2	−71.7 ± 13.7	<0.0	−98.5	−45.0	−9.1 ± 2.5	<0.0	−14.0	−4.2	0.9 ± 0.2	<0.0	0.5	1.3
Lact. 2	2.7 ± 7.7	0.7	−124	17.7	4.4 ± 14.7	0.8	−24.5	33.2	−5.4 ± 2.7	<0.0	−10.6	−0.2	0.1 ± 0.2	0.8	−0.4	0.5
Lact. ≥ 3	0 ^b^	-	-	-	0 ^b^	-	-	-	0 ^b^	-	-	-	0 ^b^	-	-	-
January	−6.5 ± 5.0	0.2	−16.4	3.4	−30.2 ± 10.2	<0.0	−50.1	−10.2	−5.2 ± 1.8	<0.0	−8.7	−1.7	0.1 ± 0.2	0.6	−0.3	0.5
February	−29.6 ± 5.7	<0.0	−40.8	−18.5	−18.1 ± 11.4	0.1	−40.4	4.2	−10.6 ± 2.0	<0.0	−14.5	−6.7	−0.5 ± 0.2	<0.0	−1.0	−0.1
March	−15.5 ± 6.3	<0.0	−27.9	−3.1	−45.8 ± 12.6	<0.0	−70.6	−21.1	−14.1 ± 2.2	<0.0	−18.4	−9.7	−0.0 ± 0.2	1.0	−0.5	0.5
April	−16.3 ± 6.6	<0.0	−29.3	−3.2	−33.6 ± 13.1	<0.0	−59.4	−7.9	−11.3 ± 2.3	<0.0	−15.8	−6.7	−0.1 ± 0.2	0.6	−0.6	0.4
May	−6.4 ± 6.8	0.4	−19.8	7.0	−58.3 ± 13.4	<0.0	−84.7	−32.0	−15.2 ± 2.4	<0.0	−19.9	−10.5	0.2 ± 0.3	0.4	−0.3	0.7
June	−9.6 ± 7.8	0.2	−24.8	5.6	−52.3 ± 15.3	<0.0	−82.3	−22.3	−23.4 ± 2.7	<0.0	−28.7	−18.1	0.6 ± 0.3	0.1	−0.1	1.0
July	−51.7 ± 8.8	<0.0	−69.0	−34.5	−57.2 ± 17.5	<0.0	−91.5	−23.0	−18.7 ± 3.1	<0.0	−24.7	−12.6	−0.6 ± 0.3	0.1	−1.2	0.1
August	−39.6 ± 7.5	<0.0	−54.3	−24.9	−46.2 ± 14.9	<0.0	−75.4	−17.1	−13.8 ± 2.6	<0.0	−19.0	−8.7	−0.3 ± 0.3	0.3	−0.8	0.3
September	−28.4 ± 6.8	<0.0	−41.9	−15.2	−48.1 ± 13.5	<0.0	−74.5	−21.8	−1.9 ± 2.4	0.4	−6.6	2.7	0.0 ± 0.3	1.0	−0.5	0.5
October	−29.3 ± 5.7	<0.0	−40.5	−18.1	−61.1 ± 11.5	<0.0	−83.7	−38.6	4.0 ± 2.1	<0.0	0.1	8.0	−0.1 ± 0.23	0.7	−0.6	0.4
November	−23.1 ± 4.5	<0.0	−32.0	−14.2	−19.8 ± 9.2	<0.0	−37.9	−1.7	−5.8 ± 1.6	<0.0	−8.9	−2.6	−0.3 ± 0.2	0.2	−0.7	0.1
December	0 ^b^	-	-	-	0 ^b^	-	-	-	0 ^b^	-	-	-	0 ^b^	-	-	-
Day −14	24.3 ± 5.5	<0.0	13.8	34.7	164.6 ± 11.0	<0.0	142.9	186.3	13.2 ± 1.9	<0.0	9.5	16.9	−1.0 ± 0.3	<0.0	−1.5	−0.5
Day −13	−4.8 ± 5.3	0.4	−15.2	5.7	128.7 ± 11.0	<0.0	107.1	150.2	11.3 ± 1.9	<0.0	7.6	15.0	−1.1 ± 0.3	<0.0	−1.6	−0.6
Day −12	−5.7 ± 5.3	0.3	−16.1	4.7	121.3 ± 11.0	<0.0	99.8	142.8	12.0 ± 1.9	<0.0	8.4	15.7	−1.0 ± 0.3	<0.0	−1.5	−0.6
Day −11	−4.5 ± 5.3	0.4	−14.9	5.9	116.4 ± 11.0	<0.0	94.9	137.9	9.8 ± 1.9	<0.0	6.1	13.5	−1.0 ± 0.3	<0.0	−1.5	−0.6
Day −10	−3.6 ± 5.3	0.5	−14.0	6.8	121.9 ± 11.0.	<0.0	100.4	143.4	9.7 ± 1.9	<0.0	6.0	13.4	−1.1 ± 0.3	<0.0	−1.6	−0.6
Day −9	−8.4 ± 5.3	0.1	−18.8	2.0	124.1 ± 11.0	<0.0	102.6	145.6	8.7 ± 1.9	<0.0	5.0	12.4	−1.1 ± 0.3	<0.0	−1.6	−0.6
Day −8	−7.0 ± 5.3	0.2	−17.4	3.4	123.2 ± 11.0	<0.0	101.7	144.7	10.2 ± 1.9	<0.0	6.5	13.9	−1.1 ± 0.3	<0.0	−1.6	−0.6
Day −7	−9.6 ± 5.3	0.1	−19.9	0.8	120.1 ± 11.0	<0.0	98.6	141.6	8.9 ± 1.9	<0.0	5.2	12.6	−1.1 ± 0.3	<0.0	−1.6	−0.6
Day −6	−1.7 ± 5.3	0.8	−12.0	8.7	107.6 ± 11.0	<0.0	86.2	129.1	7.9 ± 1.9	<0.0	4.2	11.6	−0.9 ± 0.3	<0.0	−1.4	−0.4
Day −5	−4.2 ± 5.3	0.4	−14.5	6.2	119.2 ± 11.0	<0.0	97.7	140.6	7.7 ± 1.9	<0.0	4.0	11.4	−0.9 ± 0.3	<0.0	−1.4	−0.5
Day −4	1.3 ± 5.3	0.8	−9.0	11.7	96.8 ± 10.9	<0.0	75.4	118.3	4.6 ± 1.9	<0.0	0.9	8.3	−0.8 ± 0.3	<0.0	−1.3	−0.3
Day −3	2.0 ± 5.3	0.7	−8.4	12.3	88.3 ± 11.0	<0.0	66.8	109.8	4.8 ± 1.9	<0.0	1.2	8.5	−0.6 ± 0.3	<0.0	−1.1	−0.1
Day −2	7.6 ± 5.3	0.2	−2.8	17.9	86.7 ± 11.0	<0.0	65.2	108.2	5.7 ± 1.9	<0.0	2.0	9.4	−0.5 ± 0.3	<0.0	−1.0	−0.0
Day −1	25.1 ± 5.3	<0.0	14.7	35.5	104.1 ± 11.0	<0.0	82.5	125.6	2.3 ± 1.9	0.2	−1.4	6.0	−0.2 ± 0.3	0.4	−0.7	0.3
Day 0	111.0 ± 5.0	<0.0	101.2	120.8	94.5 ± 10.4	<0.0	74.2	114.8	0.2 ± 1.8	0.9	−3.3	3.7	1.8 ± 0.2	<0.0	1.4	2.3
Day 1	146.3 ± 5.1	<0.0	136.4	156.3	−54.1 ± 10.5	<0.0	−74.6	−33.5	7.2 ± 1.8	<0.0	3.7	10.7	4.1 ± 0.2	<0.0	3.7	4.6
Day 2	130.1 ± 5.1	<0.0	120.2	140.0	−51.5 ± 10.5	<0.0	−72.0	−31.0	7.8 ± 1.8	<0.0	4.3	11.3	3.5 ± 0.2	<0.0	3.1	4.0
Day 3	96.7 ± 5.1	<0.0	86.8	106.6	−21.2 ± 10.4	<0.0	−41.7	−0.8	12.4 ± 1.8	<0.0	8.9	15.9	2.0 ± 0.2	<0.0	1.6	2.5
Day 4	66.0 ± 5.0	<0.0	56.1	75.9	−0.0 ± 10.4	1.0	−20.5	20.5	12.4 ± 1.8	<0.0	8.8	15.9	1.1 ± 0.2	<0.0	0.6	1.6
Day 5	39.6 ± 5.0	<0.0	29.7	49.5	3.7 ± 10.4	0.7	−16.8	24.1	13.3 ± 1.8	<0.0	9.7	16.8	0.6 ± 0.2	<0.0	0.1	1.0
Day 6	24.3 ± 5.0	<0.0	14.4	34.2	−11.3 ± 10.4	0.3	−31.7	9.2	11.1 ± 1.8	<0.0	7.6	14.6	0.5 ± 0.2	<0.0	0.0	1.0
Day 7	24.1 ± 5.0	<0.0	14.2	34.0	−15.1 ± 10.4	0.1	−35.6	5.3	8.4 ± 1.8	<0.0	4.9	12.0	0.5 ± 0.2	<0.0	0.1	1.0
Day 8	23.7 ± 5.0	<0.0	13.8	33.5	−11.3 ± 10.4	0.3	−31.8	9.1	7.6 ± 1.8	<0.0	4.1	11.1	0.5 ± 0.2	<0.0	0.0	1.0
Day 9	28.0 ± 5.0	<0.0	18.1	37.9	−14.5 ± 10.4	0.2	−34.9	6.0	7.2 ± 1.8	<0.0	3.7	10.8	0.6 ± 0.2	<0.0	0.1	1.0
Day 10	21.6 ± 5.0	<0.0	11.8	31.5	−5.3 ± 10.4	0.6	−25.8	15.1	5.7 ± 1.8	<0.0	2.2	9.2	0.3 ± 0.2	0.3	−0.2	0.7
Day 11	20.7 ± 5.0	<0.0	10.8	30.6	−4.7 ± 10.4	0.6	−25.2	15.7	1.8 ± 1.8	0.3	−1.7	5.3	0.5 ± 0.2	<0.0	0.0	1.0
Day 12	9.4 ± 5.0	0.1	−0.5	19.3	0.3 ± 10.4	1.0	−20.2	20.7	1.6 ± 1.8	0.4	−1.9	5.2	0.6 ± 0.2	<0.0	0.2	1.1
Day 13	10.1 ± 5.0	<0.0	0.2	20.0	3.0 ± 10.4	0.8	−17.4	23.5	2.5 ± 1.8	0.2	−1.0	6.0	0.1 ± 0.2	0.6	−0.4	0.6
Day 14	0 ^b^	-	-	-	0 ^b^	-	-	-	0 ^b^	-	-	-	0 ^b^	-	-	-

b: This coefficient is set to zero and used as a reference for comparison with other groups, as it is redundant in the analysis.

**Table 4 animals-14-01834-t004:** Classification machine learning models’ predictions of calving day using daily behavioral data for 14 days prepartum in dairy cows ^1^.

Machine Learning Models	Sensitivity (%)	Specificity (%)	Positive Predictive Values (%)	Negative Predictive Value (%)	Accuracy Score (%)	F2 Score (%)
Random forest	40.0	98.8	68.8	96.2	95.2	43.7
Decision tree	49.1	94.1	35.5	96.6	91.4	45.6
Gradient boosting	34.6	98.8	55.9	95.8	94.3	37.4
Naïve Bayes	49.1	95.6	42.1	96.6	92.7	47.5
Neural network (multilayer perceptron)	40.0	98.9	71.0	96.2	95.3	43.8

^1^ Machine learning models were developed randomly, allocating 80% of the data, and tested on the remaining 20% of the data (*n* = 298 calving). Sensitivity = TP/TP + FN; specificity = TN/TN + FN; positive predictive value = TP/TP + FP; negative predictive value = TN/TN + FN; accuracy score = TP + TN/Total predictions; F2 score = (1+ 2^2^) × Precision × Sensitivity/(2^2^ × Precision + Sensitivity); and precision = TP/(TP + FP). Here, TP = true positive, TN = true negative, FP = false positive, and FN = false negative. The AfiTag-II biosensor (Afikim Ltd., Kibbutz, Afikim, Israel) was used to capture and classify postural behavioral metrics such as activity (minutes/day), rest time (minutes/day), rest per bout (minutes/day), and restlessness ratio. These variables were used to develop machine learning models.

## Data Availability

The data and model were not deposited in an official repository. Data are available upon request to the corresponding author.

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
