# Peer review of "Behavioral Adaptations in Tropical Dairy Cows: Insights into Calving Day Predictions"

_animals, 2024, doi:10.3390/ani14121834_

Round 1

Reviewer 1 Report

Comments and Suggestions for Authors

General Comments:

The manuscript by Raza et al. outlines how on-animal accelerometer-based data can be used to classify and categorise cow behaviour over the peripartum period. By use of machine learning techniques, it also attempts to use these data to predict the day of calving. Overall it is a well-constructed paper with some interesting and useful results: the application of new technologies, both hardware and software, to address welfare issues is particularly welcome.

However there is one area I would like to raise in order to gain the most from the study. Currently, the machine leaning model used to predict calving day, and as described, a correct classification is only considered if the model predicts on the actual day of calving (which is why sensitivity for that event is generally low I would day, at ~ 50%). It would be worth considering test performance for a window of ± 1 day or ± 2 days within calving day, or maybe up to 1 day, or up to 2 days prior to caving. I suppose this will depend on the particular goals, i.e. how important is it to predict the precise day of calving. Of course the sensitivity will naturally increase, but this is something that should at least be considered in the discussion if not included in the results, although that would be relatively easy to do so.

The following is a list of points I noted as I read through the manuscript. Most are minor editorial but some will take a bit more consideration.

Specific Comments

Line 29: for “568.76 ± 5.42 minutes/day”, please indicate if it is mean ± SD or mean ± SE, that will then assist in subsequent references throughout the manuscript.

Lines 89-90: From my understanding, THI is not expressed as a percentage.

Lines 126-127: These metrics should be defined, particularly “activity rest”, or provide a reference.

Lines 137-138: I am not sure why generalised linear mixed models (GLMM) were used, rather than just linear mixed models (LMM) as presumably the data were assumed to come from a normal distribution, i.e. not binary, count etc., which is why a GLMM is used. Nonetheless, the LMM is just a special case of the GLMM.

Lines 151-153: It would have been better to make your pairwise comparisons within your final multivariable model, rather than via separate one-way ANOVAs, but I won’t require you to do so.

Line 166: Please change “guthub” to “github” in the web link. However, on checking, there is currently no content added in the Github repository.

Line 191: I think 1 decimal point accuracy for percentages is sufficient, suggest reduce from 2 d.p. to 1 d.p. throughout the manuscript.

Lines 240, 242: In Figure 1, it is not clear why these are labelled “mean differences”, not just “means”. What are they the difference of? Similarly for captions in Figures 2-8. One other issue: the responses for each parity show similar patterns, i.e. parallel. Did you fit Day × Lactation interactions in your models? This is important as the shape of the time course may differ significantly between parities, for example, maybe the older cows will show a quicker return to pre-partum levels than parity 1 cows. Similarly for Figures 2-4.

Line 352, Figure 5: are these values over the entire peri-partum period, i.e. pre-partum and post-partum combined? Similarly for Figures 6-8.

Line 440: You state that for random forest, “These findings demonstrate its effectiveness in both capturing true-calving events…” but the sensitivity is only 40%.

Lines 449-452, Table 1: Need to define “F2”.

Lines 720-721: Change “unsing” to “using’, “attacthment” to “attachment”, “surroudning” to “surrounding”.

Author Response

Respected Reviewer,

Thank you very much for your comments and suggestions about our manuscript. We have addressed the comments and updated our manuscript according to suggestions. Additionally, we are resolute to provide any further clarification and explanation required for our analysis. We have attached specific details about the comments and suggestions in the file below.

Thank you very much

Chaidate Inchaisri

Reviewer 2 Report

Comments and Suggestions for Authors

The paper titled "Behavioral adaptations in tropical dairy cows: Insights into calving day predictions" aims to tackle the challenge of monitoring activity patterns and predicting calving days for dairy cows in tropical environments. By utilizing activity-behavioral data from 289 pre-calving and 347 post-calving Holstein Friesian cows, and applying machine learning algorithms, the study demonstrates that first-lactation cows exhibit significantly reduced rest time and increased activity levels. The machine learning models, particularly random forest and gradient boosting, show promising performance in calving day predictions. These findings suggest significant implications for improving herd management and animal welfare on tropical dairy farms.

The paper aligns well with the scope of the journal, addressing important aspects of animal behavior and welfare in the context of agricultural practices.

The manuscript provides valuable insights into dairy cow behavior in tropical climates, addressing a significant gap in existing research. The use of machine learning for calving prediction is innovative and relevant, potentially enhancing dairy farm management and animal welfare. However, the paper has several shortcomings that need to be addressed.

The main question addressed is how to accurately predict calving days in tropical dairy cows using behavioral data and machine learning algorithms.

The topic is both original and relevant, filling a specific gap related to dairy cow management in tropical climates.

The study adds to the subject area by providing a novel approach to calving prediction using machine learning, tailored to tropical conditions.

There are minor typos and phrasing issues throughout the manuscript. For instance, on line 22, "calving days for dairy cows during the periparturient period on tropical dairy farms" could be rephrased for clarity. Careful proofreading is recommended to correct these errors.

Simple Summary:

The simple summary should be rewritten to provide a clearer, non-technical overview of the study's context and importance. Avoid technical jargon and nonstandard acronyms to make it accessible to non-experts.

Abstract:

The abstract correlates well with the manuscript content, but it could include more detailed results and the significance of the findings. The abstract needs to be rewritten to include more specific results and highlight the significance of the data obtained.

Keywords:

Avoid using terms in the keywords that are already present in the article title to enhance searchability.

Introduction:

lines 39 – 47 considering to cite 10.3168/jdsc.2020-0074 as example of the technology used in dairy farming

Methods

The authors should consider incorporating more diverse environmental and physiological variables to improve the robustness of the predictive models. Additional controls for environmental factors like temperature and humidity variations could be beneficial.

Report more details regarding the animals used and the diet, provide feed analysis and see 10.3390/vetsci10090554

Consider providing more detailed information on the machine learning models used, including any hyperparameter tuning and validation methods.

Report the statistical model used

Results:

very nice graphs, considering to report also some tables

Discussion:

The discussion section could benefit from a more thorough comparison with similar studies in temperate and subtropical climates, highlighting the unique challenges and findings of the tropical context.

The authors should explore the integration of additional behavioral and physiological data, such as rumination patterns and body temperature, to enhance the predictive accuracy of their models. Future research could also examine the applicability of these findings to different breeds and farming systems.

I recommend incorporating a discussion paragraph highlighting the significance of educating future veterinarians, technicians, and farmers about the issues addressed in the paper. Emphasizing the importance of effective teaching methods in shaping knowledgeable students and proficient veterinarians would add depth to the paper's implications. It is advisable to refer to recent publications on veterinary education to provide up-to-date insights into best practices in preparing future professionals to address the challenges discussed in the paper. Please see: 10.1016/j.jevs.2023.104537 and 10.3390/ani13223503.

Report also economical evaluation of the findings of the paper

Conclusion:

The conclusions are generally consistent with the evidence presented but could benefit from a more detailed discussion on the limitations and potential biases in the data.

Reference

The references appear appropriate, but a more comprehensive review of recent literature on machine learning applications in animal behavior monitoring is recommended.

Please double-check the reference list to ensure that all cited references are included in the main text and vice versa. This will help maintain the manuscript's integrity and academic rigor.

Author Response

Respected Reviewer,

Thank you very much for your insightful comments and feedback about our work. We have addressed your comments and incorporated the suggestions. If any further suggestions and explanations are required for our analysis, we will appreciate your feedback. We have attached specific details about the suggestions and feedback in the file below.

Thank you

Chaidate Inchaisri,

Round 2

Reviewer 2 Report

Comments and Suggestions for Authors

The authors have diligently addressed the review comments, significantly enhancing the paper's quality. As a result, it is now well-suited for publication